

# Transplantation of adipose-derived stem cells overexpressing inducible nitric oxide synthase ameliorates diabetes mellitus-induced erectile dysfunction in rats

Yan Zhang[1,2], Jun Yang[1,2], Li Zhuan[3], Guanghui Zang[4], Tao Wang[1,2] and Jihong Liu[1,2]

[1] Department of Urology, Tongji Hospital, Tongji Medical College, Huazhong University of Science and Technology, Wuhan, Hubei, China
[2] Institute of Urology, Tongji Hospital, Tongji Medical College, Huazhong University of Science and Technology, Wuhan, Hubei, China
[3] Department of Reproductive Medicine, the First People's Hospital of Yunnan Province, Kunming, Yunnan, China
[4] Department of Urology, Xuzhou Central Hospital, Xuzhou, Jiangsu, China

Corresponding author
Jihong Liu, jhliu@tjh.tjmu.edu.cn

## ABSTRACT

**Background:** Erectile dysfunction is a major complication of diabetes mellitus. Adipose-derived stem cells (ADSCs) have attracted much attention as a promising tool for the treatment of diabetes mellitus-induced erectile dysfunction (DMED). Inducible nitric oxide synthase (iNOS) plays an important role in protecting penile tissues from fibrosis. The aim of this study was to determine the efficacy of ADSCs overexpressing iNOS on DMED in rats.

**Methods:** ADSCs were isolated and infected with adenovirus overexpressing iNOS (named as ADSCs-iNOS). The expression of iNOS was detected using western blot analysis and real-time PCR. Rats were randomly assigned into five groups: control group, DMED group, ADSCs group, ADSCs-EGFP group and ADSCs-iNOS group. $5 \times 10^5$ cells were given once via the intracorporal route. Two weeks after treatment, erectile function was assessed by electrical stimulation of the cavernous nerve. Penile tissues were obtained and evaluated at histology level.

**Results:** We found that ADSCs-iNOS had significantly higher expression of iNOS at mRNA and protein levels and generated more nitric oxide (NO). ADSCs-iNOS reduced collagen I and collagen IV expression of corpus cavernosum smooth muscle cells (CCSMCs) in cell co-culture model. Transforming growth factor-β1 expression in CCSMCs reduced following co-culture with ADSCs-iNOS. Injection of ADSCs-iNOS significantly ameliorated DMED in rats and decreased collagen/smooth muscle cell ratio of penile tissues. Moreover, elevated NO and cyclic guanosine monophosphate concentrations were detected in penile tissues of ADSCs-iNOS group.

**Conclusion:** Taken together, ADSCs-iNOS significantly improved erectile function of DMED rats. The therapeutic effect may be achieved by increased NO generation and the suppression of collagen I and collagen IV expression in the CCSMCs to decrease penile fibrosis.

## INTRODUCTION

Erectile dysfunction (ED) is one of the most significant complications in men with diabetes mellitus (DM). Numerous studies have found that more than 50% of men with DM are afflicted with ED (*Thorve et al., 2011*). Men with DM tend to suffer from ED about 10 years earlier than the general population (*Johannes et al., 2000*). Diabetes mellitus-induced erectile dysfunction (DMED) is severe and has a negative effect on the quality of life. Phosphodiesterase 5 inhibitors are the predominant medicines to treat ED. However, diabetic male patients are less responsive to phosphodiesterase 5 inhibitors (*Cheng, 2007*). Therefore, developing new therapeutic options for DMED is urgent.

Over the past years, stem cell therapy has produced positive effects on DMED in experimental animals. Adipose-derived stem cells (ADSCs) are attracting considerable attention because they can be isolated through standardized procedures and vastly expanded, and they secrete a broad range of trophic factors (*Sowa et al., 2012*). Moreover, ADSC transplantation seems to be safe and avoid the risks of tumorigenesis (*Alagesan & Griffin, 2014*). Autologous ADSCs significantly improved erectile function of type-II diabetic rats (*Garcia et al., 2010*). Stem cell therapy has beneficial effects to increase smooth muscle and endothelium, nitric oxide (NO)-cyclic guanosine monophosphate (cGMP) pathway, neuronal nitric oxide synthase (nNOS)-positive nerve fibers, and decrease fibrosis and apoptosis in the penis (*Chen et al., 2017*; *Qiu et al., 2011*; *Ryu et al., 2012*; *Sun et al., 2012*; *Zhang et al., 2016*).

NO is a neurotransmitter which plays an important role in penile erection. NO is synthesized by three nitric oxide synthase (NOS) isoforms, including nNOS, endothelial NOS (eNOS) and inducible NOS (iNOS). Among these synthases, iNOS can produce NO continuously independent of $Ca^{2+}$ (*Eissa et al., 1998*). Recent evidence suggests that the pathogenesis of DMED may be due to impaired NO production, and iNOS acts an essential part in protecting penile tissues from the pro-fibrotic effects of hyperglycemia (*Ferrini et al., 2010*). Continuous NO production by iNOS inhibits collagen synthesis, directly reacts with ROS to produce peroxynitrite, and downregulates transforming growth factor-β1 (TGF-β1)/Smad pathway, thus reducing fibrosis level (*Gonzalez-Cadavid & Rajfer, 2010*). Gene transfer of iNOS DNA decreased TGF-β1 and plasminogen-activator inhibitor-1 expression, and regressed Peyronie's disease-like plaque (*Davila et al., 2004*). Our previous study showed that saRNA mediated iNOS expression improved DMED via endogenously generating NO (*Wang et al., 2013*).

Given the beneficial effects of ADSCs and iNOS in the treatment of ED, we hypothesized that ADSCs combined with iNOS could improve erectile function of DMED rats. In this study, we aimed to determine the efficacy of ADSCs combined with iNOS for improving erectile function in DMED rats.

## MATERIALS AND METHODS

### Rat ADSCs isolation and culture

Rat ADSCs were isolated as previously described (*Wang et al., 2015*). Briefly, adipose tissues were incised from inguen, and digested at 37 °C in 0.1% collagenase type I (Sigma-Aldrich, St. Louis, MO, USA) with vigorous shaking for 90 min. The digested tissues were

mixed with Dulbecco's modified Eagle's medium (DMEM) supplemented with 10% fetal bovine serum (FBS), and centrifuged at 220$g$ for 10 min. The cells were suspended in DMEM containing 10% FBS and filtered through a 75-μm cell striner (Solarbio, Beijing, China). Then cells were plated and cultured in DMEM containing 10% FBS at 37 °C in humidified incubator with 5% $CO_2$. Cells were passaged with trypsin/EDTA (Beyotime, Nantong, China) as required. The fourth-passage ADSCs were used in this study.

## Flow cytometry

ADSCs were characterized by flow cytometry at passage 4. Briefly, ADSCs were harvested and washed twice with phosphate-buffered saline (PBS). Then, the cells were incubated for 30 min in PBS containing anti-CD29-FITC (cat. # 555005; BD, San Diego, CA, USA), CD31-PE (cat. # 555027; BD, San Diego, CA, USA), CD49-FITC (cat. # 557457; BD, San Diego, CA, USA), CD90-PE (cat. # 551401; BD, San Diego, CA, USA), CD106-PE (cat. # 559229; BD, San Diego, CA, USA), CD34-FITC (cat. # sc-7324; Santa Cruz, Dallas, TX, USA), CD45-FITC (cat. # MCA43FT; AbD, Oxford, UK), CD73-FITC (cat. # bs-23233R; Bioss, Beijing, China), CD105-FITC (cat. # bs-10662R; Bioss, Beijing, China). The stained cells were then subjected to flow cytometry analysis.

## Differentiation of ADSCs

For adipocyte differentiation, ADSCs were cultured at a density of $1 \times 10^4$ cells/cm$^2$ in DMEM containing 10% FBS, 50 mg/L indomethacin, 0.5 mmol/L isobutylmethylxanthine, 10 mg/L insulin and one mmol/L dexamethasone (all from Sigma-Aldrich, St. Louis, MO, USA). After 21 days of induction, the cells were stained using Oil Red-O staining solution.

For smooth muscle cell differentiation, ADSCs were cultured at a density of $1 \times 10^4$ cells/cm$^2$ in low-glucose DMEM containing 10% FBS, 50 μg/mL platelet-derived growth factor-BB and five μg/L TGF-β1 (all from PeproTech, Rocky Hill, NJ, USA). After 14 days of induction, the cells were stained with the antibody against α-smooth muscle actin (α-SMA, cat. # BM0002; Boster, Wuhan, China). Immunofluorescence staining was performed as previously described (*Zhang et al., 2016*).

## Transfection with adenovirus

The fourth-passage ADSCs were cultured for 24 h. ADSCs were incubated with adenovirus expressing EGFP or iNOS-EGFP (GenecChem, Shanghai, China) at a multiplicity of infection of 30. After incubation for 12 h, the medium was changed into fresh growth medium.

## Real-time PCR

Real-time PCR was performed in cultured cells on the 1st, 3rd, 5th, 7th, 10th and 14th day after infection with adenovirus. RNA was extracted with a multisource RNA miniprep kit (Corning, NY, USA). Total RNA (500 ng) was reversely transcribed into cDNA using the PrimeScript$^{TM}$ RT reagent kit (TaKaRa, Dalian, China). Real-time PCR for each sample was performed on an MX3000P quantitative PCR system (Agilent, Santa Clara, CA, USA) using SYBR Premix Ex Taq (TaKaRa, Dalian, China) and the following primers: iNOS_ 5′-AAGCACATTTGGCAATGGAGAC-3′,

β-actin_ 5′-GACGGTGTGCACCAACATCTA-3′, 5′-TTCTTGGCTTTCAGGATGGAG-3′. PCR condition included 95 °C for 30 s, followed by 40 cycles of 95 °C for 5 s, 60 °C for 30 s and 72 °C for 30 s. Relative expression level of target genes was calculated with $2^{-\Delta\Delta Ct}$ method. β-actin was chosen as normalization control.

## Western blot analysis

Protein extraction from rat penile tissues and cultured cells was performed using NP-40 lysis buffer (Beyotime, Nantong, China). Protein concentration was detected with the BCA protein assay kit (Beyotime, Nantong, China). A total of 50 μg protein was electrophoresed on 12% sodium dodecyl sulfate/polyacrylamide gels and then blotted on a polyvinylidene difluoride membrane (Millipore, Billerica, MA, USA). The membrane was blotted with primary antibodies against collagen I (cat. # 14695-1-AP; Proteintech, Wuhan, China), collagen IV (cat. # 55131-1-AP; Proteintech, Wuhan, China), iNOS (cat. # sc-7271; Santa Cruz, Dallas, TX, USA), TGF-β1 (cat. # ab92486; Abcam, Cambridge, UK) followed by HRP-labelled secondary antibodies. The enhanced chemiluminescence detection system (Pierce; Thermo Fisher, Rockford, IL, USA) was used to detect protein bands. Each band was quantified by densitometry with Image J software.

## Measurement of NO concentration

Cells were incubated in culture medium containing 10 mmol/L L-Arginine for 24 h on the 1st, 3rd, 5th, 7th, 10th and 14th day after infection with adenovirus. Then the supernatant was harvested. NO concentrations in penile tissues or cultured cells were determined with total NO assay kit (Beyotime, Nantong, China) according to the manufacturer's instructions.

## cGMP concentration determination

The cGMP concentrations in penile tissues or cultured cells were measured using an enzyme-linked immunosorbent assay kit (Westang, Shanghai, China) according to the manufacturer's instructions.

## Rat corpus cavernosum smooth muscle cell isolation and culture

Rat corpus cavernosum smooth muscle cells (CCSMCs) were obtained from corpus cavernosum tissues as described previously with minor modifications (Chen et al., 2011). Briefly, corpus cavernosum tissues were harvested and cut into fragments of one mm$^3$. These fragments were placed in an overturned 25 cm$^2$ culture flask. Then, four mL DMEM containing 10% FBS was added into culture flask. After 30 min, the flask was flipped over. Differential adherence method was used to purify CCSMCs. Immunostaining with an antibody against α-SMA was performed to identify CCSMCs. Immunofluorescence staining was performed as previously described (Zhang et al., 2016).

## Transwell co-culture of ADSCs and CCSMCs

ADSCs and CCSMCs were co-cultured in medium containing 30 mmol/L glucose in 0.4 μm pore size Transwell inserts (Corning, NY, USA). CCSMCs were cultured in 12-well plates at $1 \times 10^4$ cells/well. ADSCs, ADSCs-EGFP or ADSCs-iNOS were seeded

on Transwell inner membrane at a density of $1 \times 10^4$ cells/well. The co-culture system allowed ADSCs and CCSMCs to grow in the same medium without direct contact between them.

## Animal experiments

Animal experiments were carried out according to the guidelines and regulations by the Ethical Committee of Tongji Hospital, Tongji Medical College, Huazhong University of Science and Technology. This study was approved by the Ethical Committee (TJ-A20131213). A total of 60 male SD rats (8-week old) were purchased from Hunan SJA Laboratory Animal Co., Ltd. After an overnight fast, 51 of them were randomly selected and received intraperitoneal injection with streptozotocin (60 mg/kg, Sigma-Aldrich, St. Louis, MO, USA) (Yang et al., 2013). The remaining nine rats were used as controls. Three days after streptozotocin injection, rats with fasting blood glucose level higher than 16.7 mmol/L were selected as DM. After 8 weeks, DM rats were selected by the apomorphine (APO) test. Briefly, APO (80 mg/kg, Sigma-Aldrich, St. Louis, MO, USA) was injected in the loose skin of neck. DMED rats were identified if rats did not exhibit an erectile response within 30 min.

DMED rats were randomly divided into different treatment groups (nine rats in each group): DMED group received no treatment; ADSCs group received corpus cavernosum injection of 60 μL PBS containing $5 \times 10^5$ ADSCs; ADSCs-EGFP group received corpus cavernosum injection of 60 μL PBS containing $5 \times 10^5$ ADSCs-EGFP; ADSCs-iNOS group received corpus cavernosum injection of 60 μL PBS containing $5 \times 10^5$ ADSCs-iNOS.

## Assessment of erectile function

After 2 weeks, intracavernous pressure (ICP) and mean arterial pressure (MAP) were detected to evaluate erectile function as described previously (Yang et al., 2013). The stimulus parameters were two voltages (2.5, 5.0 volts), frequency 15 Hz, duration 1 min and pulse width 1.2 ms. Erectile response was measured at 2.5 and 5.0 voltages. Pressure was measured and recorded with a data acquisition system (AD Instruments Powerlab/4SP, Bella Vista, NSW, Australia). Then, penile tissues were harvested for measurement of NO and cGMP concentration ($n$ = 4/group) and masson trichrome staining ($n$ = 5/group).

## Masson trichrome staining

Penile tissues were embedded in paraffin and cut into five μm thick sections. Masson trichrome staining was performed as previously described (Yang et al., 2013). Image Pro Plus 6.0 software (Media Cybernetics Inc, Bethesda, MD, USA) was used to quantitatively analyze smooth muscle content and collagen in five randomly selected specimens per group.

## Statistical analysis

Results were expressed as mean ± standard deviation. Data were analyzed with one-way analysis of variance followed by Tukey-Kramer test for post hoc comparisons using

GraphPad Prism 5.0 (GraphPad Software, San Diego, CA, USA). Normal distribution was determined by Kolmogorov–Smirnov test. $P < 0.05$ was considered statistically significant.

## RESULTS

### Isolation and characterization of ADSCs

The isolated ADSCs exhibited spindle-shaped morphology at passage 1 (Fig. 1A). ADSCs expanded in vitro and showed fibroblast-like shape at passage 4 (Fig. 1B). To characterize ADSCs, surface markers of the cultured cells at passage 4 were determined using flow cytometry. As shown in Figs. 1C–1K, most of the cells expressed CD29 and CD90, while few cells were positive for CD34, CD45, CD73, CD105, CD31 and CD106. To identify the multi-lineage differentiation ability of ADSCs, we induced adipogenic and myogenic differentiation of ADSCs for 21 or 14 days in appropriate induction medium. Oil Red-O staining indicated that ADSCs could differentiate to adipocytes (Fig. 1L). Immunofluorescence staining with an antibody against α-SMA indicated that ADSCs could differentiate to smooth muscle cells (Fig. 1M).

After culture for 5 to 7 days primary rat CCSMCs in spindle shape grew out of corpus cavernosum tissues. After about 2 weeks of culture, cells were passaged and purified by the differential adherence method (Fig. 1N). The cells were identified as CCSMCs by immunostaining of α-SMA (Fig. 1O).

### Overexpression of iNOS in ADSCs

To overexpress iNOS in ADSCs, we infected the cells with recombinant adenovirus. A total of 3 days after infection, fluorescence microscopy was used to observe EGFP-expressing ADSCs (Figs. 2A–2C). Real-time PCR and Western blot analysis showed that the mRNA and protein expression levels of iNOS were significantly higher in ADSCs-iNOS group than ADSCs-EGFP and ADSCs groups ($P < 0.05$, Figs. 2D and 2E). The expression of iNOS in ADSCs persisted for up to 14 days.

### Effects of iNOS overexpression on CCSMCs

As expected, NO concentration in supernatant of ADSCs-iNOS was significantly higher than that of ADSCs and ADSCs-EGFP ($P < 0.05$). Moreover, NO concentration in supernatant of ADSCs-iNOS reached maximum level on the 7th day ($P < 0.05$, Fig. 3A). We detected the effect of ADSC-iNOS on CCSMCs using Transwell co-culture model. The expression of collagen I and collagen IV in CCSMCs significantly increased in high glucose concentration ($P < 0.05$). However, significant reduction of collagen I and collagen IV expression was observed in CCSMCs co-cultured with ADSCs-iNOS ($P < 0.05$, Figs. 3B–3D). In addition, we found that TGF-β1 expression in CCSMCs reduced following co-culture with ADSCs-iNOS ($P < 0.05$, Fig. 3E).

### Effects of ADSCs-iNOS on erectile function

Finally we evaluated the effects of ADSCs-iNOS on erectile function of DMED rats. There was no significant difference in initial body weight or serum glucose concentration among the five groups of rats. A total of 14 days after the induction of diabetes, body weight was significantly lower and fasting glucose concentration was significantly higher in the

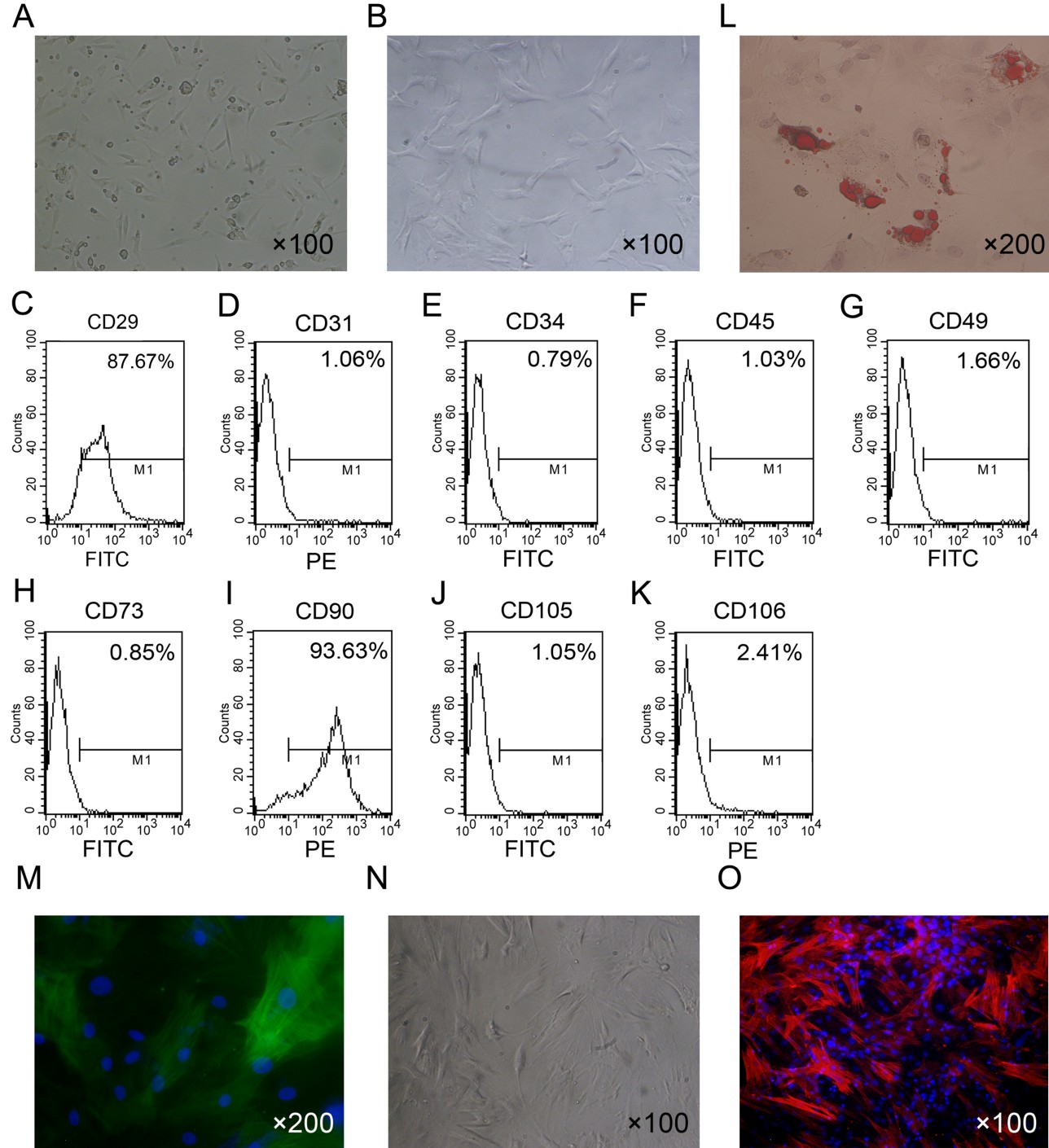

**Figure 1 Primary culture and characterization of rat ADSCs and CCSCMs.** (A) Morphological features of ADSCs at passage 1 (primary magnification: ×100). (B) Morphological features of ADSCs at passage 4 (primary magnification: ×100). (C–K) ADSCs were identified by flow cytometry at passage 4. (L) Adipogenic differentiation of ADSCs assessed by Oil Red-O staining (primary magnification: ×200). (M) Myogenic differentiation of ADSCs assessed by immunofluorescence staining with α-SMA antibody (green, primary magnification: ×200). (N) Morphological features of CCSMCs after purification (primary magnification: ×100). (O) CCSMCs were assessed by immunostaining with α-SMA antibody (red, primary magnification: ×100). The nuclei were labeled with DAPI (blue).

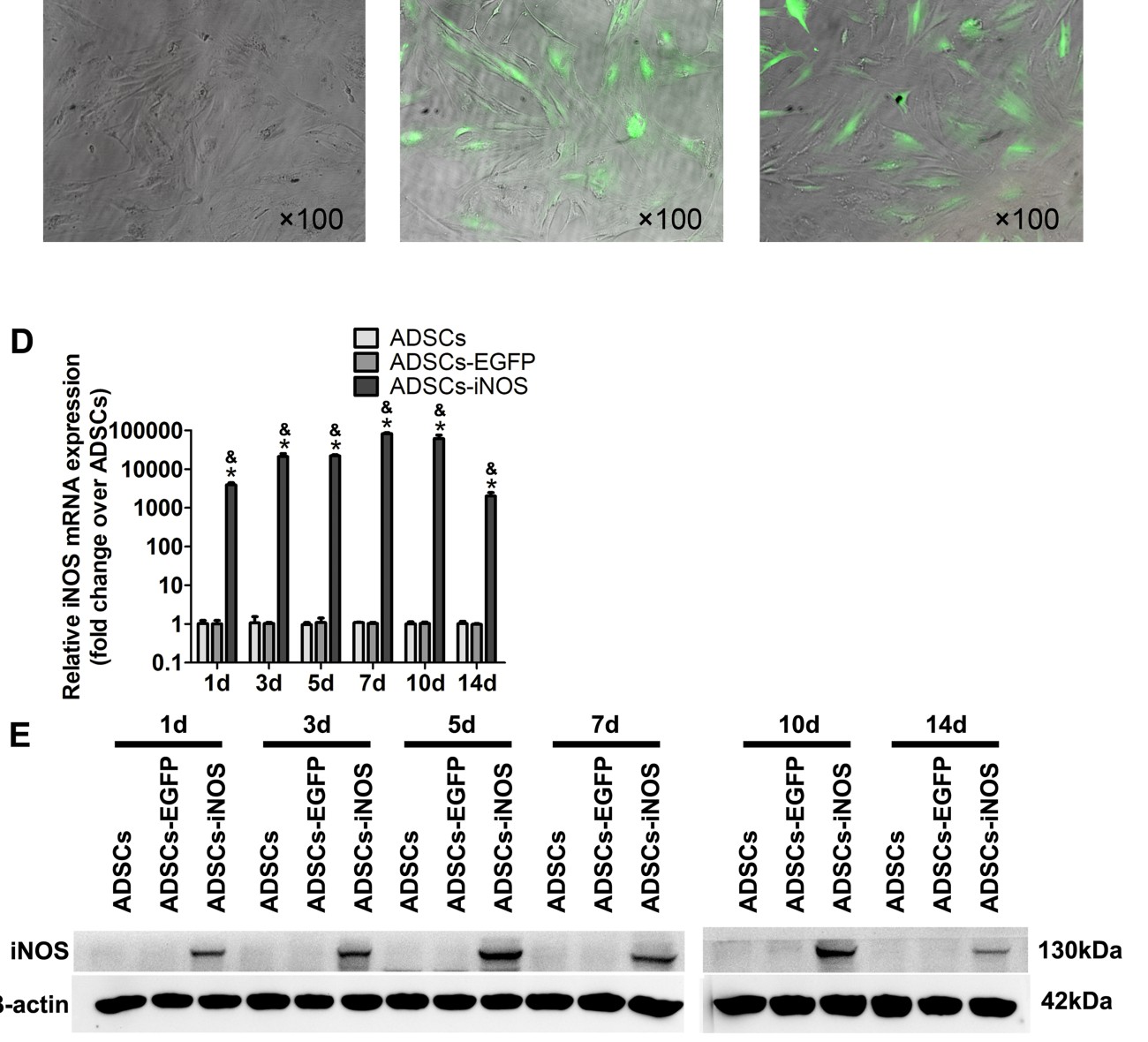

**Figure 2 iNOS expression in ADSCs.** (A–C) EGFP-expressing ADSCs were observed by fluorescence microscopy 3 days after infection (primary magnification: ×100). (D) iNOS expression at mRNA level was detected by real-time PCR and the relative ratio of iNOS/β-actin measured in ADSCs was arbitrarily presented as 1. Data are the mean of three independent experiments. (E) iNOS expression at protein level was detected by Western blot analysis. Data are the mean of three independent experiments. $^*P < 0.05$ vs. ADSCs, $^\&P < 0.05$ vs. ADSCs-EGFP.

diabetic rats than the age-matched control rats ($P < 0.05$). However, there were no significant differences in body weight or serum glucose concentration among DMED, ADSCs, ADSCs-EGFP and ADSCs-iNOS groups (Table 1).

Erectile function was measured 2 weeks after ADSCs transplantation. The ICP/MAP ratio was significantly lower in DMED group than that in control group. Intracavernous injection of ADSCs or ADSCs-EGFP significantly improved erectile function of DMED

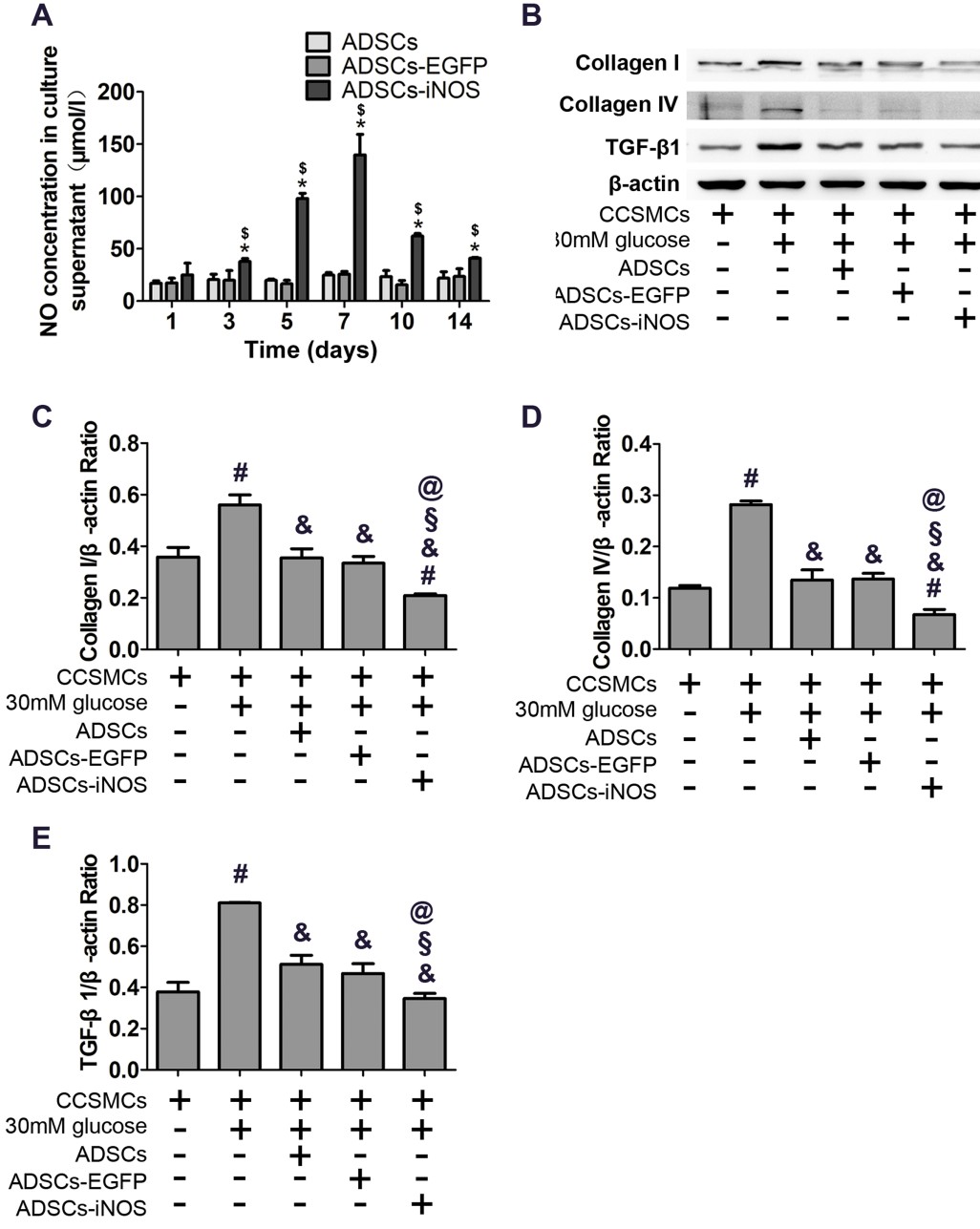

**Figure 3 Characterization of ADSCs-iNOS.** (A) NO concentration in supernatant of ADSCs was measured after incubation in 10 mmol/L L-Arginine for 24 h. Data are the mean of three independent experiments. (B) The protein expression of collagen I, collagen IV and TGF-β1 in CCSMCs was detected by Western blot analysis after 7 days of co-culture. (C) Data were shown as the relative density values of collagen I to β-actin as loading control. Data are the mean of three independent experiments. (D) Data were shown as the relative density values of collagen IV to β-actin as loading control. Data are the mean of three independent experiments. (E) Data were shown as the relative density values of TGF-β1 to β-actin as loading control. Data are the mean of three independent experiments. $^{*}P < 0.05$ vs. ADSCs, $^{\$}P < 0.05$ vs. ADSCs-EGFP, $^{\#}P < 0.05$ vs. CCSMCs, $^{\&}P < 0.05$ vs. CCSMCs cultured in DMEM containing 30 mM glucose, $^{\S}P < 0.05$ vs. CCSMCs co-cultured with ADSCs, $^{@}P < 0.05$ vs. CCSMCs co-cultured with ADSCs-EGFP.

**Table 1  Metabolic and physiological variables of experimental rats.**

| Variable | Control | DMED | ADSCs | ADSCs-EGFP | ADSCs-iNOS |
|---|---|---|---|---|---|
| Initial weight (g) | 268.0 ± 12.7 | 268.9 ± 8.6 | 268.4 ± 7.9 | 266.6 ± 12.9 | 270.7 ± 11.3 |
| Final weight (g) | 596.0 ± 54.5 | 274.8 ± 18.1[*] | 284.5 ± 22.3[*] | 281.8 ± 17.3[*] | 285.1 ± 18.8[*] |
| Initial fasting glucose (mmol/L) | 6.3 ± 0.5 | 6.2 ± 0.3 | 6.2 ± 0.4 | 6.2 ± 0.5 | 6.2 ± 0.4 |
| Final fasting glucose (mmol/L) | 6.2 ± 0.3 | 30.8 ± 2.0[*] | 29.4 ± 2.1[*] | 30.3 ± 1.6[*] | 30.0 ± 2.8[*] |
| MAP (mmHg) | 104.4 ± 5.7 | 105.1 ± 7.7 | 105.8 ± 6.3 | 102.0 ± 7.7 | 108.4 ± 6.4 |

Notes:

DMED, diabetes mellitus-induced erectile dysfunction; ADSCs, adipose-derived stem cells; iNOS, inducible nitric oxide synthase; MAP, mean arterial pressure.

[*] $P < 0.05$ vs. control group. Data expressed as mean ± standard deviation.

rats. The ICP/MAP ratio of ADSCs-iNOS group was markedly elevated compared to DMED, ADSCs and ADSCs-EGFP groups (all $P < 0.05$, Figs. 4A and 4B). In addition, NO and cGMP concentrations in penile tissues of ADSCs and ADSCs-EGFP groups were significantly elevated compared to DMED group, although they were still significant lower compared to ADSCs-iNOS group ($P < 0.05$, Figs. 4C and 4D). The mean collagen/smooth muscle cell ratio significantly increased in the DMED rats compared to the control rats ($P < 0.05$). Intracavernous injection of ADSCs or ADSCs-EGFP attenuated collagen content ($P < 0.05$). Furthermore, mean collagen/smooth muscle cell ratio in ADSCs-iNOS treated rats was significantly lower than that in ADSCs or ADSCs-EGFP group ($P < 0.05$) (Figs. 4E–4J).

# DISCUSSION

In this study, we demonstrated that infection of ADSCs with adenovirus containing iNOS expression cassette led to significantly high expression of iNOS and increased generation of NO. ADSCs played a positive role in restoring DMED in rats. Furthermore, overexpression of iNOS in ADSCs was shown to achieve a significantly larger improvement of erectile function. The therapeutic effect may be achieved by increased NO generation and the suppression of collagen I and collagen IV expression in the CCSMCs to decrease penile fibrosis.

Corpus cavernosum is composed of a loose trabecular meshwork of smooth muscle and connective tissues, which are structural basis of erectile function. *Ryu et al. (2004)* found that collagen fiber content was significantly increased in corpus cavernosum of patients with vasculogenic ED. Further studies demonstrated that extracellular matrix such as fibronectin, collagen IV and collagen I accumulated and collagen I/collagen III ratio was decreased in the corpus cavernosum tissue of diabetic ED rats (*Hirata et al., 2009*; *Zhou et al., 2012*). Abnormal extracellular matrix can cause mechanical alterations of corpus cavernosum, which may provoke penile venous leakage, leading to vasculogenic ED (*Li et al., 2013*). Increased expression of TGF-β1 pathway could be involved in collagen fiber accumulation and penile fibrosis (*Ryu et al., 2004*). CCSMCs are the prdominant mesenchymal cell type in the corpus cavernosum. Synthesis of connective tissue proteins and collagen by CCSMCs was significantly increased by exogenous TGF-β1 (*Moreland et al., 1995*). In accordance with previous studies, our study showed that collagen I and collagen IV expression in CCSMCs significantly increased under high glucose condition

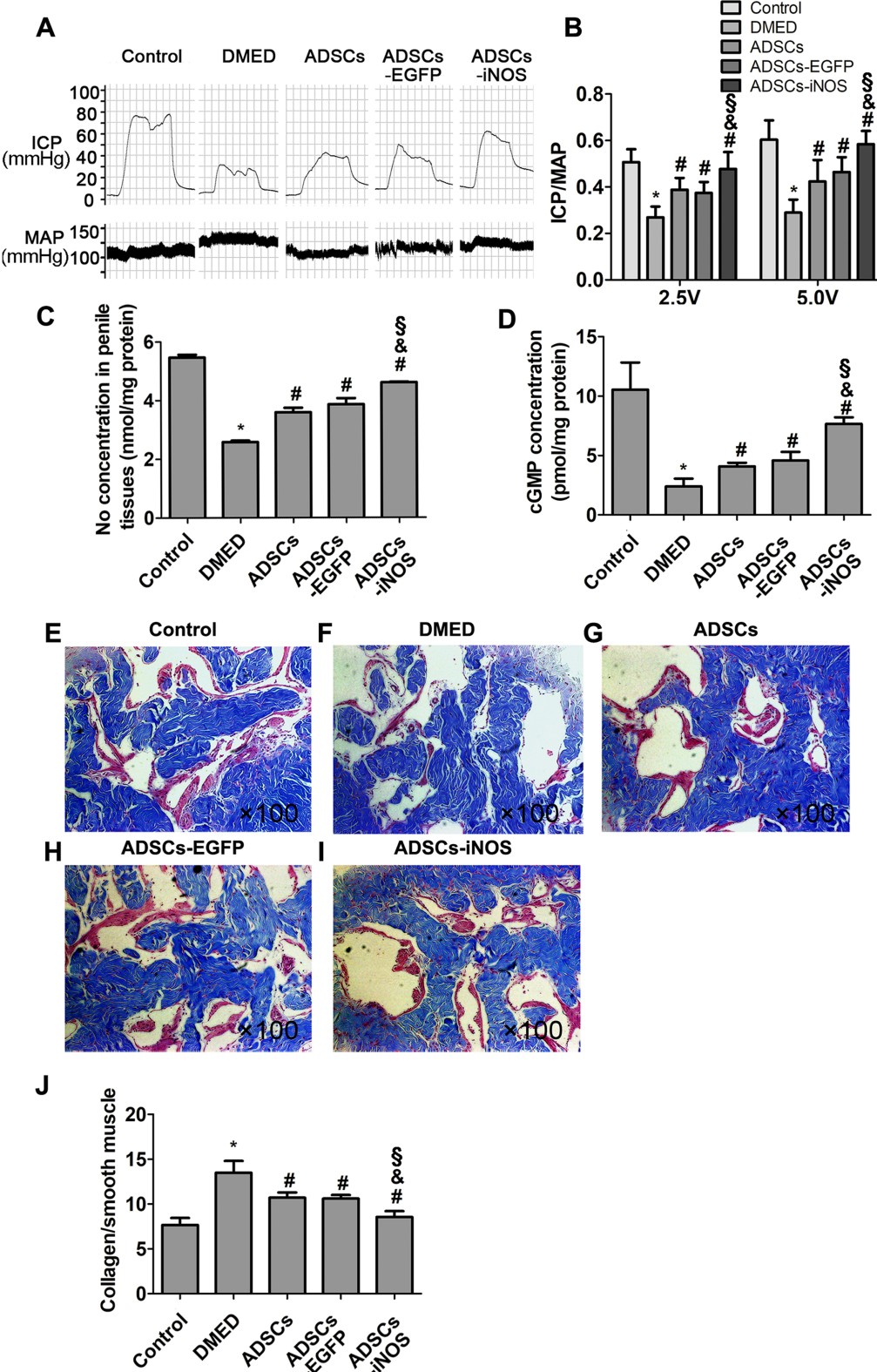

**Figure 4 The transplantation of ADSCs-iNOS improved erectile function of DMED rats.** (A) MAP and ICP response to electrostimulation of cavernous nerves (5 volts, 1 min). (B) ICP/MAP ratio in control, DMED, ADSCs, ADSCs-EGFP and ADSCs-iNOS groups. (C) NO concentration in penile tissues

**Figure 4** (continued)
of each group. Data are the mean of four rats per group. (D) ELISA assay of cGMP concentration in penile tissues of each group. Data are the mean of four rats per group. (E–I) Penile tissues were stained with Masson trichrome in all groups. Collagen fibers were stained blue, while smooth muscle was stained red (primary magnification: ×100). (J) The collagen to smooth muscle ratio in penile tissues of each group. Data are the mean of five rats per group. $^*P < 0.05$ vs. control group, $^\#P < 0.05$ vs. DMED group, $^\&P < 0.05$ vs. ADSCs group, $^\S P < 0.05$ vs. ADSCs-EGFP. 

and penile fibrosis significantly increased in DMED rats. Then we co-cultured CCSMCs with ADSCs, and found significantly decreased synthesis of collagen I and collagen IV. Furthermore, we found that TGF-β1 expression in CCSMCs significantly decreased, indicating the involvement of TGF-β1 signaling pathway. However, further studies are needed to elucidate precise mechanism of ADSCs' effect on CCSMCs. In vivo, ADSCs ameliorated penile fibrosis in ED rats, and the possible mechanism is still needed to be elucidated precisely. ADSCs significantly prevented upregulation of collagen III and elastin in the tunica albuginea of Peyronie's disease rats (*Castiglione et al., 2013*). Intratunical injection of ADSCs decreased the expression of tissue inhibitors of metalloproteinases, and promoted the expression and activity of matrix metalloproteinases (*Gokce et al., 2014*). In addition, exosomes from ADSCs were enriched with miR-132 and miR-let7 which played antifibrotic role through regulating TGF-β1 pathway (*Zhu et al., 2017*).

NO/cGMP signaling pathway plays a crucial role of in erectile function. NO released from nonadrenergic noncholinergic nerves and endothelium is the principal mediator of penile erection. Moreover, this pathway plays an important role in ameliorating tissue fibrosis. *Saura et al. (2005)* found that NO inhibited TGF-β/Smad induced gene transactivation in a cGMP-dependent manner in endothelial cells, leading to proteasomal degradation of phosphorylated Smad. In addition, NO reduced and delayed nuclear translocation of activated Smad (*Saura et al., 2005*). cGMP significantly reduced TGF-β induced upregulation of collagen I and collagen II at mRNA and protein levels (*Beyer et al., 2015*). In this study, we found that iNOS overexpression in ADSCs significantly enhanced NO generation and could last for 14 days. Compared to ADSCs, ADSCs-iNOS significantly increased NO and cGMP concentrations in penile tissues, and decreased collagen I and collagen IV expression in CCSMCs to ameliorate penile fibrosis in DMED rats. These results are consistent with previous studies that the inhibition of iNOS promoted penile fibrosis. For example, Monica et al found that more collagen was deposited in corpus cavernosum of iNOS knockout mouse. In addition, inactivation of iNOS gene led to exacerbated penile fibrosis in DM mouse by increasing oxidative stress and TGF-β1 expression under hyperglycemia condition (*Ferrini et al., 2010*). Similarly, L-NIL as an inhibitor of iNOS activity increased oxidative stress and fibrosis in the media of arteries (*Ferrini et al., 2004*).

However, other studies suggested that erectile function was impaired by iNOS expression. The induction of iNOS accompanying penile fibrosis was demonstrated in rat models of DMED, Peyronie's disease and aging-related ED (*Bivalacqua et al., 2000*; *Ferrini et al., 2001*; *Usta et al., 2003*). In addition, iNOS inhibition improved erectile function in

Peyronie's disease or DMED rat. However, the application of aminoguanidine as an iNOS inhibitor was not appropriate, because aminoguanidine inhibited not only all three kinds of NOS, but also advanced glycation end products involved in tissue fibrosis. Moreover, further studies are needed to determine if elevated iNOS expression is possibly a compensatory reflection when eNOS and nNOS expression is decreased under pathological conditions.

There are some limitations in the current study. First, we did not investigate the effect of iNOS expression on the stemness of ADSCs. In our study, more NO was released from ADSCs-iNOS than ADSCs. *Tapia-Limonchi et al. (2016)* found that NO maintained embryonic stem cells (ESC) pluripotency and delayed ESC differentiation by regulating Gsk3-β/β-catenin and PI3K/Akt signaling pathways. The effect of NO released from ADSCs iNOS on ADSC stemness required further investigation. Second, ADSCs were only transplanted into corpus cavernosum once. Repeated injection of ADSCs may achieve better therapeutic effects but need to be confirmed in future studies. Third, we did not compare therapeutic effect of ADSCs-iNOS to that of iNOS alone. Although adenovirus act as efficient agents for gene transfer, they can activate cellular and humoral immune response in the hosts, which limits the safety and efficacy in vivo. The innate immune response is mediated by the adenovirus particle, but not viral transcription (*Muruve, 2004*). In contrast, transplanted stem cells could suppress excessive immune response (*Lin, Lin & Lue, 2012*). ADSCs overexpressing iNOS may be safer and more efficient than iNOS alone. Fourth, the statistical power for this study was relatively low (49.9%) which may be due to the small group size, and more experimental rats will be included to validate these results in our further study. Fifth, our study did not detect distinct downstream pathways of TGF-β1 to determine the precise mechanism of ADSCs-iNOS' effect on CCSMCs. A separate but more extensive experiment on this topic will be carried out.

## CONCLUSIONS

In conclusion, intracavernous administration of ADSCs-iNOS improved erectile function of DMED rats. Injection of ADSCs-iNOS in a rat model of DM significantly decreased penile fibrosis, possibly due to increased NO generation and suppressed expression of collagen I and collagen IV in the CCSMCs.

### Funding

The work was funded by the National Natural Science Foundation of China (grant no. 81070484). The funders had no role in study design, data collection and analysis, decision to publish, or preparation of the manuscript.

### Grant Disclosure

The following grant information was disclosed by the authors:
National Natural Science Foundation of China: 81070484.

## Competing Interests

The authors declare that they have no competing interests.

## Author Contributions

- Yan Zhang performed the experiments, analyzed the data, contributed reagents/materials/analysis tools, prepared figures and/or tables, authored or reviewed drafts of the paper, approved the final draft.
- Jun Yang performed the experiments, contributed reagents/materials/analysis tools, authored or reviewed drafts of the paper, approved the final draft.
- Li Zhuan performed the experiments, analyzed the data, contributed reagents/materials/analysis tools, prepared figures and/or tables, approved the final draft.
- Guanghui Zang performed the experiments, contributed reagents/materials/analysis tools, approved the final draft.
- Tao Wang conceived and designed the experiments, approved the final draft.
- Jihong Liu conceived and designed the experiments, approved the final draft.

## Animal Ethics

The following information was supplied relating to ethical approvals (i.e., approving body and any reference numbers):

The Ethical Committee of Tongji Hospital, Tongji Medical College, Huazhong University of Science and Technology approved the animal experiments of this study (TJ-A20131213).

## Data Availability

The raw measurements are available in the Supplemental Files.

## Supplemental Information

Supplemental information for this article can be found online at http://dx.doi.org/10.7717/peerj.7507#supplemental-information.

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
