# Peer review of "Transplantation of adipose-derived stem cells overexpressing inducible nitric oxide synthase ameliorates diabetes mellitus-induced erectile dysfunction in rats"

_PeerJ, doi:10.7717/peerj.7507_

## Round 0.1 · original submission · Major Revisions

Both reviewers and I found this to be an interesting manuscript which makes an interesting contribution to the field. However, reviewer-1 in particular raised some valid concerns regarding the immunoblots of phospho-Smad which are valid criticisms and which need to be addressed. Those issues aside, the rest of the comments largely should be relatively easy to address. I look forward to seeing the revised manuscript.

Reviewer 1 ·

Basic reporting

There are a few minor typographical errors:
Line 57 should read "secrete" rather than "secret"
Line 58 should read "ADSC" rather than "ADSCs"
Line 62 should read "in the penis" rather than "in penis"
Line 110 should read "fourth" rather than "forth"
Line 312 should read "penile" rather than "penil"

Experimental design

There are a few elements of the methods and materials that are missing such that the experiments would be impossible to replicate as it stands: These are detailed below:

The number of passages through which the ADSCs have been cultured should be given for all experiments, at the moment it is only stated for some experiments.

The catalogue numbers of all of the antibodies used in the FACs, immunofluorescence microscopy and immunoblotting experiments should be provided.

It is not clear if the adenovirus expressing iNOS-EGFP expresses a fusion protein or iNOS and EGFP from separate promoters, as through most of the results it is annotated as ADSCs-iNOS and statistical comparisons should only really be made with statistical tests between ADSCs expressing EGFP and not ADSCs alone.

There is no mention of how the immunofluorescence microscopy experiments were performed.

It is not clear how the immunoblotting was performed - what secondary antibodies were used and how were they detected? From the blots provided in the raw data it would appear to be enhanced chemiluminescence with HRP-labelled secondary antibodies, but this is not stated.

It should also be noted that the CCK-8 assay essentially measures levels of reduced NADH and NADPH, so it is a measure of metabolic activity and viability, not a measure of proliferation as the authors state.

Validity of the findings

In Figure 2A, the authors provide images of EGFP-adenovirus-infected cells, but not iNOS-EGFP-adenovirus-infected cells, which is equally important to assess infection efficiency.

As mentioned above, the appropriate statistical comparison is between ADSCs expressing EGFP and those expressing iNOS in Figures 2B, 2D, 3B, 3D-G, 4B-D. These data should not be compared with ADSCs alone, as any difference could be ascribed to viral infection and/or expression of iNOS.

The number of independent replicates undertaken for all of the data needs to be clear in the figure legends of each data figure.

The raw data for Figure 3A does not appear to have been provided.

The raw data tiff files provided for the immunoblots for Figure 2-2 are over-exposed and new data should be provided.

The immunoblots for p-Smad2/3 and Smad2/3 exhibit multiple bands, indicating a great deal of non-specific binding of the antibody. Do the authors have evidence that the bands they have quantified are actually Smad2/3? As a consequence the immunoblots are not of a quality suitable for quantification and more clear immunoblots should be provided. With any improved immunoblots, quantification should examine Smad2 and Smad3 individually, otherwise any data will underestimate potential differences.

How were the immunoblotting experiments performed to give the cell culture protein expression? The ratio will vary with the exposure of each immunoblot and unless all replicates were immunoblotted at exactly the same time, probed with secondary antibodies at the same time and imaged at the same time. Therefore it seems unlikely the data can be presented without being normalised to one condition for each experiment (as it is in Figure 1C, for example).

Additional comments

The manuscript is clearly written and convincing yet there are issues I have highlighted with respect to the detail of the methods and materials used and the quality of the immunoblots. Furthermore, without better labelling it is difficult to tell whether the supplemental files of the raw immunoblot data are all of the replicates or just some of them. I would also point out that lines 289-293 in the conclusions indicate one particular mechanism by which the ADSCs may signal to the CCSMCs in co-culture, but there are many alternatives to miRNAs in exosomes and this should be discussed in further detail.

·

Basic reporting

This a clearly written, logically structured manuscript that succinctly described the results of experiments designed to test the potential therapeutic utility of adipose derived stem cells over expressing iNOS in experimental diabetic erectile dysfunction. The background literature is well summarised and supports the authors novel and testable hypothesis.

The data figures are well constructed and presented.

Experimental design

The experiments described in this manuscript are logically designed to advance previous studies in this field; specifically the therapeutic utility of ADSC's and iNOS-mediated NO generation in DMED.

The methods employed are appropriate and span biochemical, cell biological and in vivo techniques. Taken together these experiments represent a comprehensive test of the hypothesis.

Whilst the findings from the in vivo model are entirely consistent with iNOS-mediated NO produced by transplanted ADSC's exerting beneficial effects this has not been directly verified (by the co-administration of an iNOS selective inhibitor) and therefore the authors should be cautious when discussing this point.

The authors have not quantified the number of ADSC-iNOS cells that remain the erectile tissue after transplantation. It is clear that NO and cGMP levels are elevated following transplantation but it is not clear where the transplanted cells end up.

Validity of the findings

No comment

Additional comments

This is a well written and logical manuscript that describes the results of a series of experiments testing the therapeutic utility of ADSC-iNOS in a preclinical model of DMED. The data presented broadly support the conclusions and provide novel findings that advance our understanding of the role and therapeutic utility of iNOS in ED.

---

## Round 0.2 · Minor Revisions

I am satisfied that all comments have been addressed, with two minor exceptions which I hope you can address. Both of which are, I think, issues of clarity. Please show an expanded version of the pSMAD blots in figure 3; there is a possibility that some of the pSMAD signal has been inadvertently cropped from the bottom of the figure. For clarity and transparency, please show a slightly larger version of this blot. Second, regarding the point 13 in the rebuttal, the key question raised by the reviewer (though I accept perhaps not as clearly as it could have been), is how do you control for inter-experimental variation in blots. Hence, from experiment 1, how do you compare to experiment 2? Is there a common control sample on both? If not, how do you make the comparison? Could you please make this clearer in the revised paper?

Meanwhile, thank you for a cogent and careful rebuttal of all the other points raised.

---

## Round 0.3 · accepted · Accept

Thanks for attending to these outstanding points. I am happy with your response and so recommend publication.

#